# Systematic review of the relationship between burn-out and spiritual health in doctors

Ishbel Orla Whitehead ![ORCID], Suzanne Moffatt, Stephanie Warwick ![ORCID], Gemma F Spiers ![ORCID], Tafadzwa Patience Kunonga ![ORCID], Eugene Tang, Barbara Hanratty ![ORCID]

Population Health Sciences Institute, Newcastle University, Newcastle upon Tyne, UK

**Correspondence to**
Dr Ishbel Orla Whitehead;
orla.whitehead@newcastle.ac.uk

## ABSTRACT

**Objective** To investigate the relationship between burn-out and spiritual health among medical doctors.
**Design** Systematic literature review and narrative synthesis of cross-sectional studies.
**Setting** Any setting, worldwide.
**Data sources** Five databases were searched from inception to March 2022, including Medline, Embase, PsycINFO, Scopus and Web of Science.
**Eligibility criteria** Any study design that involved medical doctors (and other healthcare staff if assessed alongside medical doctors), that measured (in any way) both burn-out (or similar) and spiritual health (or similar) medical doctors.
**Data extraction and synthesis** All records were double screened. Data extraction was performed by one reviewer and a proportion (10%) checked by a second reviewer. Quality was assessed using the Appraisal of Cross-sectional Studies tool. Due to the heterogeneity of the included studies, a narrative review was undertaken without a meta-analysis.
**Results** Searches yielded 1049 studies. 40 studies met eligibility criteria and were included in this review. Low reported levels of spirituality were associated with high burn-out scores and vice versa. Religion was not significantly associated with lower levels of burn-out. Few studies reported statistically significant findings, few used validated spiritual scores and most were vulnerable to sampling bias.
**Conclusions** Published research suggests that burn-out is linked to spiritual health in medical doctors but not to religion. Robust research is needed to confirm these findings and develop effective interventions.
**PROSPERO registration number** CRD42020200145.

## STRENGTHS AND LIMITATIONS OF THIS STUDY

⇒ A comprehensive, systematic approach was taken to searches, with expert information scientist input.
⇒ Broad search terms were used to elicit all the available evidence, as there is inconsistency in headings used in different databases, cultures and countries.
⇒ A validated tool was used for quality assessment—the Appraisal of Cross-sectional Studies tool.
⇒ The heterogeneity of measures used for burn-out and spiritual health meant that meta-analysis was not possible.

## Definitions

Spiritual health: The authors use a definition of spiritual health developed by UK general practitioners: self-actualisation and meaning; transcendence and relationships beyond the self; and expressions of spirituality. This term is used to reflect the broad and diverse experiences of medical doctors.
Burn-out: The authors use a definition in use by the WHO of an occupational phenomenon resulting in exhaustion; distance, negativity or cynicism towards the job and reduced professional efficacy.

## INTRODUCTION

Burn-out is defined by the WHO as an occupational phenomenon, resulting in exhaustion, mental distance or feelings of negativity or cynicism towards the job, and reduced professional efficacy.[1] This has long been a concern for medical doctors, and the COVID-19 pandemic has increased attention on the health and well-being of health professionals.[2 3] Medical doctors' descriptions of burn-out include a loss of meaning in work, and objectification of patients and their families, rather than engaging with their humanity.[4] As health services globally are placed under increasing pressure, there is a growing concern around physician burn-out, moral injury and related concepts causing harm to the workforce.[5 6] Burn-out leads to retirement and resignation[7] which adds to the workforce crisis, as well as patient safety concerns.[8] The COVID-19 pandemic has presented huge challenges to the healthcare workforce.[9 10] Burn-out appears to have a malign effect on all aspects of health,[11 12] as well as being impacted itself by all aspects of health.[13] The aetiology of burn-out is still

not fully understood, is likely multifactorial and merits further research.[14–16]

The biomedical model that shapes much modern medical practice, has so far been unable to offer a comprehensive and holistic understanding of the burn-out phenomenon.[17–19] This, along with the framing of burn-out as an occupational phenomenon rather than a mental illness,[1] suggests that a different lens may be needed to develop effective approaches to burn-out . In a recent survey, general practitioners (GPs) defined spiritual health in ways that overlapped with current constructs of burn-out.[12] Spiritual health was framed by medical doctors as a positive concept.[12] Spiritual health encompasses meaning, purpose, self-actualisation, transcendence and relationships with others beyond the self.[12] Spiritual or religious practices, including mindfulness and yoga, were also included in spiritual health definitions, and both are promoted as approaches to reduce burn-out in the workplace.[20 21] Burn-out has been associated with moral injury,[22] referring to the harm caused when someone is required to act contrary to their internal ethical code.[23] Being true to a personal or religious ethical code was also an aspect of spiritual health reported by GPs.[12] Some authors argue that religiosity and spirituality are one and the same,[24] whereas English GPs defined spiritual health as including, but not exclusive to, religion,[12] and there is a growth in the population that consider themselves 'spiritual but not religious.'[25] The concepts of spiritual health and burn-out have been found to overlap in other groups of professionals.[26–28] This raises the possibility that promoting spiritual health may offer an effective approach to prevention and mitigation of burn-out for medical doctors, with opportunities for intervention at an individual and organisational level.

This systematic review asks whether there is a measured association between burn-out and spiritual health among medical doctors. It will consider the quantity and quality of quantitative evidence on burn-out, spiritual health and related concepts including spiritual well-being and distress, and religiosity by comparing all studies that have performed a measure of burn-out and a measure of spiritual health, both in the broadest terms, in qualified medical doctors worldwide.

## METHODS
### Searches and selection of studies
Studies to be included in the review were identified by searching the following databases: (1) EMBASE (1974–March 2022), (2) PsycINFO (1806– March 2022), (3) Ovid MEDLINE (1946–March 2022), (4) Web of Science and (5) Scopus. The last search was performed on the 8 March 2022. The search strategy (online supplemental file 1) was developed in Embase and adapted for use in other databases by an information scientist. For comprehensiveness, both MESH headings and key terms were used. All searches were completely rerun in March 2022. A Cochrane filter[29] and term 'not editorial or letter' was added to filter out non-studies and opinion pieces. Citations from the studies found were searched for previous similar studies. The searches were peer reviewed by an information scientist at Newcastle University.

Eligible studies were those reporting quantified evidence about both burn-out (or similar concepts including occupational stress, moral injury from work, occupational compassion fatigue, etc) and spiritual health (or concepts that included this aspect of health). These data must be reported in a study population that included medical doctors from any specialty or in postgraduate training. Any study design was eligible, but commentaries and editorials were excluded. The WHO definition for burn-out[1] and the Whitehead et al[12] definitions were referred to and considered by two reviewers at screening stage. There were no language restrictions to the searches.

### Population
Medical doctors (postgraduate registered medical professionals from any specialty).

### Exposure
Spiritual health, spiritual wellness, spiritual well-being, spirituality, religiosity or similar concepts or subphenomena, as defined by doctors.[12]

### Outcome
Burn-out, moral injury, job satisfaction, compassion fatigue, work stress or similar concepts that could come within the WHO definition for burn-out.[1]

### Study design
Any study with quantification of the interventions and outcomes of interest.

Search terms were deliberately comprehensive, and related or similar terms to burn-out or spiritual health were included in the searches, for example, compassion fatigue, moral injury, mindfulness. Studies were included if we felt that the study measured any phenomenon very close to occupational burn-out, and any measure of any aspect of well-being close to spiritual health.

### Data extraction and synthesis
All records were double screened at title and abstract, with disagreements resolved by discussion with a third reviewer, or taken to full text screening. Deduplication was done in EndNote V.X9.[30] Rayyan software was used to manage records.[31] Attempts were made to retrieve all articles, with requests to authors. Emails to authors suggested anecdotally that there could be publication and availability bias, and therefore, unpublished studies, found via conference abstracts, were included, and authors were contacted for more information where limited data were presented. No other grey literature was included. Data extraction was performed by one reviewer and a proportion (10%) checked by a second reviewer. Study quality was assessed using the Appraisal of Cross-sectional Studies tool (AXIS).[32] While this tool

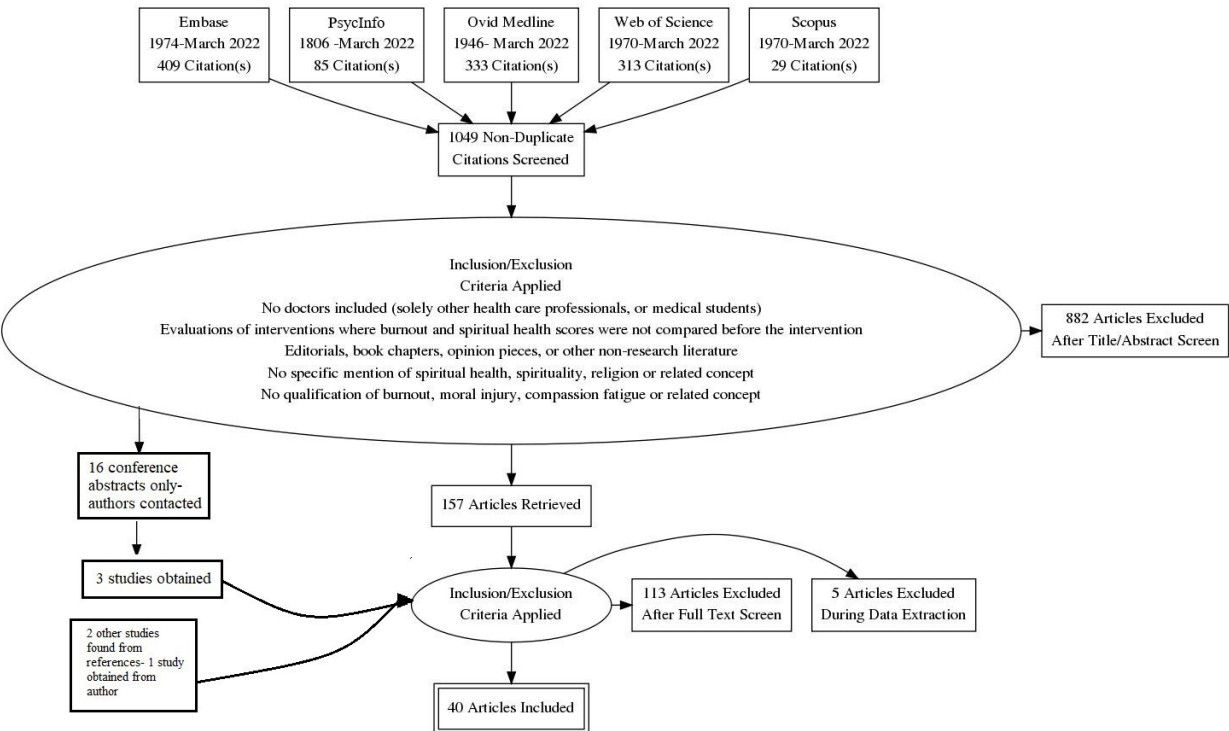

**Figure 1** PRISMA (Preferred Reporting Items for Systematic Reviews and Meta-Analyses) diagram.

does not give a 'numerical score', it assessed the methodology and internal consistency of the studies, as well as risk of bias, and as such allows assessment of the merits of each criterion as applied to that study. Heterogeneity in study design, scales and methods of analysis meant that meta-analysis was not possible, and a narrative synthesis was conducted. Subgroups were considered to see if this gave a greater sense of direction of the evidence. If statistics, for example, ORs or relative risks, were provided, these have been summarised within the narrative analysis. Custom made tables were used to analyse the assessments used for burn-out and spiritual health, and synthesised the data, including common biases.

The review was registered on Prospero, available from: https://www.crd.york.ac.uk/prospero/display_record. php?ID=CRD42020200145 and the protocol can be found in online supplemental file 2.

### Patient and public involvement

This systematic review was prompted by patient and public involvement in a previous study looking at spiritual health in primary care. The public felt that the health of medical doctors, and burn-out, was a priority for this topic.

### RESULTS

Forty studies met the inclusion criteria, from screening of the 1049 records identified in the electronic searches (figure 1). All included studies provided an assessment of burn-out and spiritual health (or related concepts) but there was a high level of heterogeneity in the measures used.

The number of study participants ranged from 4501 in the largest study[33] to 7 in the smallest.[34] Twenty studies included other healthcare staff in addition to the medical staff participants.[9 34–52] Four studies included primary care physicians,[42 53–55] with most studies involving medical doctors from intensive or critical care, emergency departments and internal medicine (figure 2).[33–36 39 46–48 50 51 53 55–64] No studies were identified including medical doctors in psychiatry or public health. Data from medical doctors were not always analysed separately.[9 35 36 38 39 44 49 51]

The oldest study was also the largest, published in 1999.[33] Three studies were conducted during the COVID-19 pandemic.[9 51 58] A US study showing lower burn-out scores in those who had a faith (including 'spiritual but not religious'),[51] a Singapore study with participants who preferred religious coping to thanks from the public[9] and a Portuguese study which showed no association between burn-out scores and religiosity in medical doctors.[58] Five of the forty studies reported findings from surveys in 2020 onwards.[9 51 56 58 65]

Almost half of the studies were from the USA[33 37 38 44 47 51–53 59–64 66–70] reflecting the dominance of North American researchers in this area (figure 3). Twelve studies included participants from Europe.[36 39 43 45 46 49 50 54 56–58 71] The largest Asian study recruited widely, pooling data from diverse countries in wealth, religion and culture, such as Saudi Arabia, Laos and Bangladesh.[48]

### Study quality

The majority of studies were assessed to be poor quality, with risks of selection bias, response bias, availability

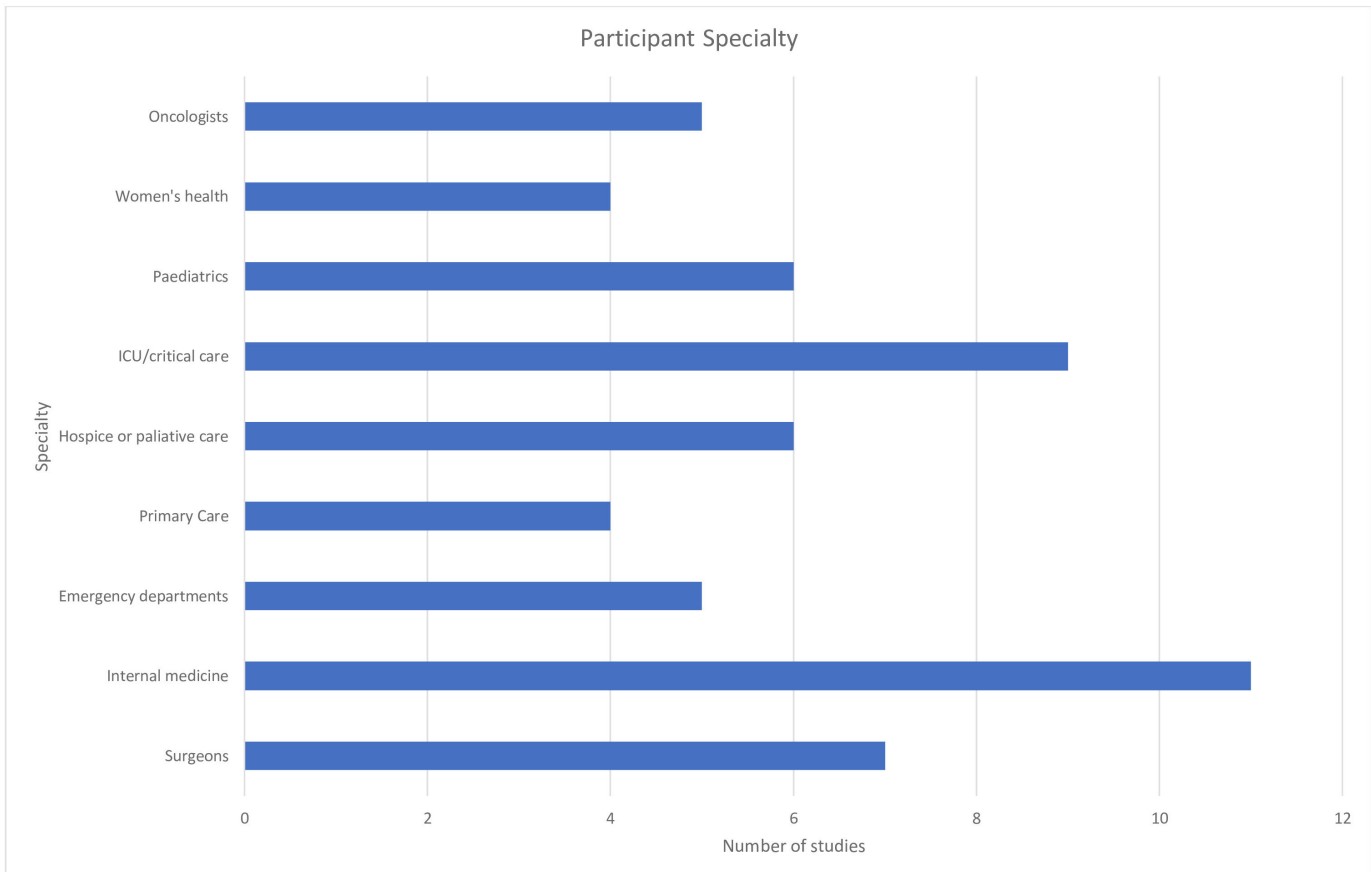

**Figure 2** Medical specialty of participants. (ICU= Intensive Care Unit)

bias and bias. One study, Roslan *et al*[55] met all the AXIS criteria fully, including justification of sample size, and there were seven studies which were of otherwise high quality as assessed by the AXIS tool (meeting 18–19 of the 20 criteria).[40–42 44 46 61 67] Three studies were not yet published in full, and therefore, not had peer review.[38 52 54] Authors reported difficulty publishing their findings.[38 72] This, as well as other studies where data were gathered on burn-out and spiritual health, but not all comparison data were published,[9 35 45 54 55 66] may have introduced availability bias. These were included as it was suspected this topic may be affected by publication bias. Most studies did not use a validated measure of spiritual health, with many studies using non-validated assessments of burn-out.

### Relationship between burn-out and spiritual health in medical doctors

These heterogeneous studies drew a wide range of conclusions (online supplemental table 1). Seventeen studies found an association between spiritual health or related concepts and lower levels of burn-out or related concepts,[37 39 40 42–45 48 51 52 54 55 57 64 69 73 74] 17 studies found no statistically significant association,[33–35 38 41 46 47 49 50 53 58–60 62 63 68 71] and 1 study found higher compassion fatigue in those using spiritual type behaviours.[61] Across the eight studies rated high in quality, evidence remained inconsistent. There was evidence of: an inverse association between religiosity and burn-out[42 44]; no association between spiritual

concepts and burn-out scores[46 67]; an association between prayer and meditation and higher compassion fatigue[61]; an association between irregular spirituality routines and burn-out[55]; a link between no spirituality and low personal accomplishment[40] and an association between spirituality and high emotional exhaustion on univariate analysis but not on multivariate analysis.[46] One study found an association between personal accomplishment and burn-out, on exclusion of respondents who omitted the spirituality question.[41] Where statistically significant associations between spiritual health (or similar), or religiosity, and burn-out (or similar) were found, the effects were typically small.

The removal of studies rated low in quality (score 12 or under on AXIS criteria) does not enhance consistency the findings. Further, there does not appear to be a pattern in terms of whether an association is shown based on country, whether that country has a state religion, but this assessment is limited by the predominance of the USA in these studies. The three most recent studies from the COVID-19 pandemic[9 51 58] do not clarify the direction of evidence, with one study finding an association,[51] one did in nurses but not medical doctors[58] and one where participants agreed they used religious coping, but did not compare religious coping and burn-out scores.[9]

Studies which looked at wider spiritual health only, rather than asking about religion, were more likely to

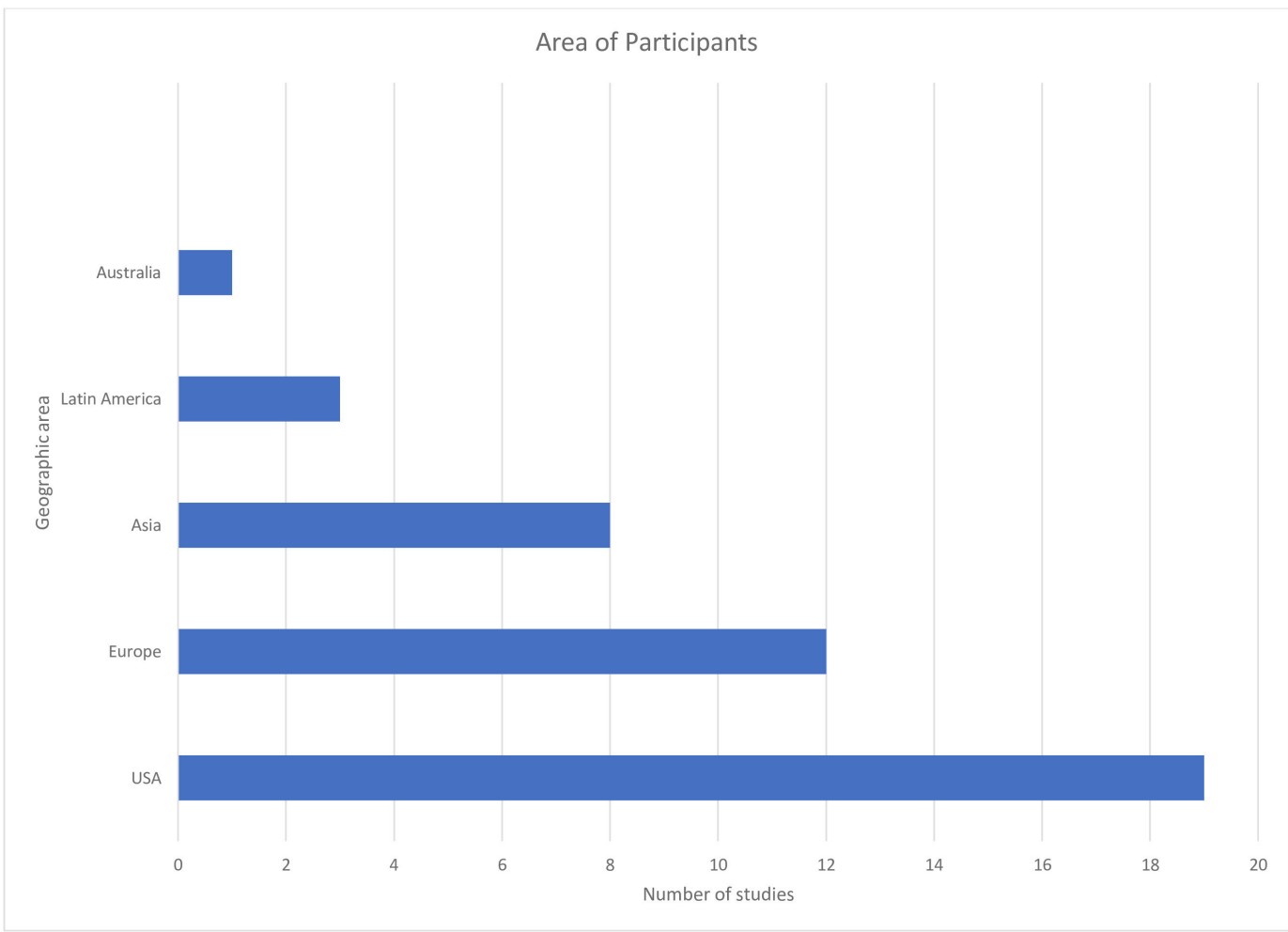

**Figure 3** Geographical location of participants of studies included in this review.

find an association between burn-out and spiritual health. Seven studies of those which asked or measure spirituality and not religion found an association,[37 39 40 43 52 57 69] and three studies did not.[36 60 63] Religion, or religious coping, alone does not appear to have a consistent effect on burn-out rates, however, wider spiritual health may have some effect.

### Quantification of burn-out

Twenty-one studies[34 35 39–41 46–48 50–53 56 57 60 64 66 67 69 71 74] used the complete Maslach Burnout Inventory (MBI)[75] (online supplemental table 1). One study used a two-item 'MBI'[62] in an emergency department population that has been found to correlate with the full MBI in a similar population.[76] Two studies used the Oldenburg Inventory,[45 58] a measure similar to MBI, but more applicable to people not directly involved in patient care. It also measures engagement with work, not including professional accomplishment.[77] Three more recent studies[9 55 63] used the Copenhagen Burnout Inventory,[78] a free to use (MBI is commercially available) measure of personal, work and patient-related burn-out, which identifies a similar burned out group to the MBI.[79] Other tools used included the Perceived Stress Scale,[42 73] Cool Down

Index[36] and Work-Related Strain Inventory.[68] While these do not measure burn-out directly, they have been found to correlate with burn-out as measured by the MBI.[36 68 80 81] The Professional quality of life (proQOLl) scale used by two studies[54 59] measures compassion fatigue, which is considered to be a phenomenon including burn-out by the ProQOL authors.[57] One study[61] used the Compassion Fatigue Test for Helpers. Clark *et al*[37] used an eight-item job satisfaction scale, which included some domains that could be considered as spiritual, for example, the sense of calling, meaning and purpose in life and being of worth.[37] Frank *et al*[33] also measured job satisfaction and desire to become a physician again. Shetach *et al*[49] also used a job satisfaction questionnaire, translated into Hebrew, designed for an unpublished thesis in marketing, and not validated with healthcare staff.

### Assessing spiritual health, well-being, spirituality and/or religiosity

An array of methods were used to assess spiritual health or similar concepts (online supplemental table 2). Validated tools include the Functional Assessment of Chronic Illness Therapy- Spiritual Well-Being Scale, a modified version for non-illness (FACIT-Sp-Non-Illness)[82]; the Jarel

spiritual well-being scale[83] and Hatch spiritual involvement and belief scale.[84] Büssing *et al*[36] used the Aspects of Spirituality questionnaire, developed by the authors in a niche setting of a particular philosophical institution.[85] The Fetzer Institute Multidimensional Measurement of Religiousness/Spirituality for Use in Health Research[86] was also used to assess religious practice, along with general, unvalidated, questions about religious commitment and spirituality.[62] A systematic review identified the FACIT and the Spirituality Index of Well-being[87] as being the highest most well validated instruments,[88] however, only one unpublished study[52] used the former, and none used the latter. Many studies used general coping scores, such as Brief COPE[89] and the Hobfoll's Questionnaire,[90] general quality of life scores or including spiritual practice in general questions, which have aspects of spiritual health as part of a wider assessment. Some studies used single questions, for example, 'my job gives me meaning',[57] 'my religious and spiritual beliefs strongly influence my work and patients',[41] 'spirituality/religion is important in my life'[64] or 'do you consider yourself spiritual'.[40 69] Seventeen studies which asked about religion only (online supplemental table 2). The Duke University Religion Index[91] measures organisational religiosity, measuring religious practice and was used alongside Hoge's Intrinsic Religiosity Scale by Ramondetta *et al*.[68] The difference between spirituality and religiosity is not always delineated by the authors: Frank *et al* used religiosity data asking how 'strongly' participants were religious, and then argue that physicians could be encouraged to explore spirituality to increase satisfaction,[33] however, spirituality is not addressed in the data, and Macuka *et al* discussed religiosity, but their question appeared to ask more generally about spirituality.[45] Ramondetta *et al* asked participants whether they were religious, spiritual, both or neither, but only quantified religiosity.[68]

## DISCUSSION
### Key findings
There is an increasing attention on both burn-out and on the relationship between medical doctors' burn-out and their spiritual health. This has become even more critical in the aftermath of the COVID-19 pandemic. Most of the research in this review reported positive associations between higher levels of spiritual health or related concepts, and lower levels of burn-out or related phenomena. A number of studies found no association, with two describing greater distress with higher levels of religiosity,[56 61] perhaps due to use of religious coping as a negative strategy.[92] Similar to other reviews in this area, research from the USA is dominant, with few studies from more secular areas. (The USA has high levels of religiosity[93 94] than, eg, many northern European countries.[95]) The one high-quality study on the topic stated that spirituality routines lower the risk of burn-out,[55] as they found that regular spiritual routines were associated with lower burn-out risk, however, we cannot infer causation

from the study design. Few studies reported statistically significant findings and the marked study heterogeneity did not support meta-analysis. While this review suggests that there is probably a link between spiritual health and burn-out in medical doctors, the quality of the evidence is poor, and high-quality robust quantitative studies of the relationship between spiritual health and burn-out in medical doctors is needed.

### Comparison with other work
To our knowledge, this is the most comprehensive review of burn-out and spiritual health in medical doctors to date. This review offers a different perspective from other work in this area that use a biomedical approach to burn-out rather than a holistic approach to include spiritual health.[96–98] Williams *et al*'s systematic review of the consequences of physician burn-out considered the physical and mental health implications of burn-out in their review, but neglected to mention spiritual health or distress.[97] Chow *et al*'s scoping review of religion and spirituality in residents included discussion of the link between reduced burn-out and spirituality.[99] Sibeoni *et al*'s systematic review and meta synthesis of physician's views of burn-out includes aspects of spiritual health in 'calmness or letting go', but did not analyse spiritual health as an aspect of overall health.[100] These authors reported that individual factors (including spiritual) and relational factors were more important than organisational concerns in protecting medical doctors from burn-out.[100] Current emphasis is on the role of COVID-19 in medical doctors' burn-out[101] with Jefferson *et al* finding studies considered gender and age effects on burn-out during COVID-19[10] but spiritual health is not mentioned. While this review is limited by the quality of the studies included, there does appear to be merit in the hypothesis of looking wider than the biomedical model.

### Common study limitations
There were several limitations common to the studies in this review: response bias, social desirability bias, sampling bias, survivorship/selection bias and cultural or language bias. There also appears to be availability and publication bias, with the topic of spiritual health being controversial—with data on spiritual health not reported in the published paper, despite being collected.[54 56] The majority of studies were vulnerable to non-response and social desirability bias. Response rates were highly variable, range 5%–99%. Use of gatekeepers, such as heads of departments[57] or directors[48 50 69] could cause sampling bias, as could use of convenience sampling at conferences.[56 59 64] Non-responders were seldom (eight studies) described.[33 36 40 41 44 55 61 65] Pressures from the COVID-19 pandemic may have affected studies carried out during this period.[9 51 58] Social desirability bias was possible due to staff feeling unable to respond honestly in very religious, or very secular environments, or share the true depth of their burn-out symptoms.[40] Studies that took place in the work setting were vulnerable to survivorship/

selection bias.[38–42 46 64] Purvis *et al* excluded residents, as they were not permanent staff.[47]

Just under half (n=19) of the studies used the MBI,[75] a measure of burn-out often considered the 'gold standard'[102] burn-out inventory. Often studies presented data on the influence of factors such as spiritual health on burn-out, when one burnout-domain only was affected.[40 41 60] The use of other inventories was not justified as to whether this was because these were better tools in their context, because they are free to use, or another reason.

Despite there being multiple available scales of religiosity or spiritual health and well-being,[88 103 104] few studies used a validated tool to assess personal spiritual health or related concepts. With the growth of the 'spiritual, but not religious'[51] and secular societies,[105] there is merit in using validated tool to assess current spiritual health state.[88] The single questions about spiritual well-being or similar were untested and unvalidated, and therefore, may have introduced information bias to the studies, and limit the utility of the data.

The use of measures in English with populations whose first language is not English may affect the quality of results. In some studies, measures were translated into the native language.[34–36 39 45 49 57 74] However, in other studies, English measures were used, without regard to the language and culture of participants,[40 48 56 73] or without stating which language/translation was used.

## Strengths and limitations of this review

This review is broad, and novel, including a wide definition of aspects of spiritual health. The lack of consistency in indexing for spiritual health and religiosity across electronic database (Medical Subject Headings) meant development of comprehensive searches was a challenge. However, the diverse range and number of papers screened suggests that the searches were sufficiently broad. Only observational, cross-sectional studies were found, and therefore, we were unable to analyse whether there is any causal relationship between spiritual health and burn-out. While most studies are from the USA, the other studies are diverse in country of setting, and cover multiple different cultures and faiths. An objective assessment was made of study quality, using the AXIS tool. Unfortunately, heterogeneity of the measures and analysis used in the studies has meant meta analysis was not possible. Some published abstracts were unobtainable, and publication bias likely affects this review, although the direction of this effect is unknown. This was mitigated by the inclusion of unpublished data on the topic.[38 52 54]

## Future research

While there have been studies that have identified this topic as being of interest, these have rarely used validated measures of both variables—burn-out and spiritual health. Research using accepted and validated tools to measure both spiritual health and burn-out are needed, especially in Northern Europe, where research is limited.

Future works should also strive to recruit study populations that are not limited to people with specific interest in the topic or strong religious views. Identification, or refutation, of a link between spiritual health and burn-out will allow fresh assessment of current organisational and individual interventions for burn-out. Developing and evaluating interventions that improve medical doctors' spiritual health may mitigate against burn-out.

## CONCLUSIONS

A link between spiritual health and burn-out in medical doctors has been hypothesised by multiple studies, and interest in the link appears to be growing. However, studies investigating this have been vulnerable to many methodological flaws. Despite this, and limited evidence for efficacy,[106] practices rooted in spiritual practice, such as mindfulness, meditation and yoga, are often recommended to prevent burn-out.[47 107 108] Since there does not appear to be a relationship between the narrower concept of religiosity and burn-out, research into any association between broader spiritual health and burn-out, using validated tools, is needed to examine the hypothesis that spiritual health and burn-out are associated concepts, and to prompt research into interventions that could prevent burn-out in medical doctors.

**Acknowledgements** Thank you to Catherine Richmond and Hannah O'Keefe for their information science input. Thank you to Carol Jagger for her input.

**Contributors** Conception and design of study:IOW, SM and BH. Search design and searching:IOW Screening records:IOW, SM, SW, GFS, ET, TPK and BH. Data curation:IOW and BH. Analysis and interpretation of data:IOW, SW and GFS. Visualisation and drafting the manuscript:IOW, SM, GFS, SW, CJ and BH. Revising the manuscript critically for important intellectual content and authorising for publication:IOW, SM, SW, BH, GFS, TPK and ET. Guarantor:OW.

**Funding** IOW is funded by the National Institute for Health Research (NIHR) on an in practice fellowship, grant number NIHR301074. BH and SM are funded by the NIHR Applied Research Collaboration (ARC) North East and North Cumbria (grant number not applicable).

**Disclaimer** The views expressed are those of the authors and not necessarily those of the NIHR or the Department of Health and Social Care.

**Competing interests** None declared.

**Patient and public involvement** Patients and/or the public were involved in the design, or conduct, or reporting, or dissemination plans of this research. Refer to the Methods section for further details.

**Patient consent for publication** Not applicable.

**Provenance and peer review** Not commissioned; externally peer reviewed.

**Data availability statement** All data relevant to the study are included in the article or uploaded as online supplemental information. All data relevant is included in the study itself. As a systematic review, the data used is in the studies referenced, or avaliable from the study's authors. Template collection forms, data extracted from included studies and used for all analyses are available on contacting the authors.

includes any translated material, BMJ does not warrant the accuracy and reliability of the translations (including but not limited to local regulations, clinical guidelines, terminology, drug names and drug dosages), and is not responsible for any error and/or omissions arising from translation and adaptation or otherwise.

**Open access** This is an open access article distributed in accordance with the Creative Commons Attribution 4.0 Unported (CC BY 4.0) license, which permits others to copy, redistribute, remix, transform and build upon this work for any purpose, provided the original work is properly cited, a link to the licence is given, and indication of whether changes were made. See: https://creativecommons.org/licenses/by/4.0/.

**ORCID iDs**
Ishbel Orla Whitehead http://orcid.org/0000-0002-4171-8583
Stephanie Warwick http://orcid.org/0000-0003-3499-7200
Gemma F Spiers http://orcid.org/0000-0003-2121-4529
Tafadzwa Patience Kunonga http://orcid.org/0000-0002-6193-1365
Barbara Hanratty http://orcid.org/0000-0002-3122-7190

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
