## [Reviewer comments · BMJ Open]

ARTICLE DETAILS

TITLE (PROVISIONAL)	A systematic review of the relationship between burnout and spiritual health in doctors
AUTHORS	Whitehead, Ishbel Orla; Moffatt, Suzanne; Warwick, Stephanie; Spiers, Gemma; Kunonga, Tafadzwa; Tang, Eugene; Hanratty, Barbara

VERSION 1 – REVIEW

REVIEWER	LUIS ÁNGEL PÉRULA DE TORRES Instituto Maimonides de Investigacion Biomedica de Cordoba
REVIEW RETURNED	31-Oct-2022

GENERAL COMMENTS	This systematic review asks whether there is a measured association between burnout and spiritual health amongst doctors. This is a qualitative systematic review, since there was no possibility of performing a meta-analysis According to the authors, mindfulness would also be included in spiritual health definitions, and both are promoted as approaches to reduce burnout in the workplace. There are numerous studies that evaluate the effect of mindfulness on doctor burnout, and these should have been taken into account in the systematic review. Although this systematic review asks whether there is a measured association between burnout and spiritual health amongst doctors (for which they look for studies with a cross-sectional design), that is, they do not try to prove the existence of a causal relationship, it would have been much more interesting to study this question and thus be able to provide evidence on whether spiritual health is a protective factor against burnout, ideally through experimental studies. One study with a very small sample size (n=7) was included in the review, probably providing hardly any valid information. The conclusions that can be reached after this systematic review are few and of low added value for clinical practice, given the study question posed (perhaps not very ambitious) and the heterogeneous and inconsistent results observed by the authors, although, given the scarcity of existing literature on this topic provides a view of the state of the issue that may be useful to implement analytical or experimental studies that allow testing hypotheses. In the discussion it is stated that "one high quality study on the topic found that spirituality routines lower the risk of burnout" and that conclusion of causality cannot be reached with a cross-sectional design such as the one mentioned.
--

REVIEWER	Francesco Chirico Università Cattolica del Sacro Cuore Facoltà di Medicina e Chirurgia
REVIEW RETURNED	28-Nov-2022

GENERAL COMMENTS	The PICO strategy and the methods are not sufficiently described to publish this paper as systematic review
---

REVIEWER	Stephanie Harris AdventHealth, Center for Nursing, Whole-Person, and Academic Research
REVIEW RETURNED	07-Dec-2022

GENERAL COMMENTS	This paper is well done, fills a gap in the literature, and is methodologically sound. Please address in the narrative whether the retrieval was limited to cross-sectional studies or whether that was the only study design retrieved, i.e. what was the rationale for including only this study design. I see that all study designs were included in the search strategy, but this should be stated in the text as well. Additionally, briefly touch on your population of interest and who your strategy included, i.e. did you include resident physicians and fellows? Physicians in all settings, specialties etc. Again, this is in the search strategy, but would be helpful in the narrative. The background/introduction should be revised to include the following additional points of discussion: the distinction/overlap between religion and spirituality, 2) etiology and ramifications of burnout in physicians (what contributes to burnout and why is burnout in physicians important?), and 3) what are some of the related concepts to burnout (i.e. compassion fatigue) that were searched and what, generally, is their relationship to burnout. One issue with this paper is the addition of moral injury into the introduction and the search strategy. Though somewhat related to burnout, it is a distinct construct with different contributing factors, and outcomes and it should not be conflated with burnout. The inclusion of this concept is somewhat problematic. As such, the relationship between moral injury, burnout, and spirituality should be explored in the introduction and in the discussion. This can also be addressed in the limitations. In addition, the authors should address the issue that religion/spirituality and spiritual well-being/health are not synonymous. Religion/spirituality may be used as a negative coping strategy. It would be helpful to distinguish those papers that distinctly address spiritual well-being versus spiritual/religious observance. Lines 105-106 unclear. Does this mean that the papers retrieved directly from authors were not peer-reviewed and thus considered grey literature? Please clarify this statement and the statement about publication bias. Additionally, was any other grey literature searched? Overall, the authors conducted a thorough systematic review and this is a valuable addition to the literature.
---

REVIEWER	Edward Spilg University of Ottawa, Department of Medicine
REVIEW RETURNED	05-Jan-2023

GENERAL COMMENTS	Overall, this is a well written manuscript which addresses an
---

	important gap in the literature. Abstract: This is clear and appropriately informative. The authors should specify in lines 29, 32, 34 (and throughout the text) which type of "doctors" ie medical doctors or consider using the term physicians to avoid confusion with other types of "doctors" given the term "doctors" is more widely applied to healthcare professionals other than physicians in North America than it is in the UK. Introduction: This clearly sets out the rationale and justification for the systematic review. This appears well written. Line 89: suggest change term doctors to physicians. Methods: As a non-expert in systematic review methodology, this section appears appropriate but I would recommend it is checked by an expert in systematic review methodology. Line 96: suggest change term doctors to physicians. Results: The results are clear and stated in a systematic fashion and focus on the study objectives. As a non-expert in systematic review methodology, I would recommend this section is checked by an expert in systematic review methodology. Lines 133, 134, 139, 182: suggest change term doctors to physicians. Table 2: suggest change term doctors to physicians throughout for consistency. Discussion: This section, I believe, is well written overall. Lines 252, 260, 261, 264, 274, 326: suggest change term doctors to physicians. Line 256-257: The statement "with a few studies from more secular areas" needs clarification, especially why the USA is not considered a "secular area" for this purposes of this study. Line 269 and 289: should replace "residents" with "resident physicians" Conclusion: The conclusions are clear and relate to the findings in the text. Lines 329, 336: suggest change term doctors to physicians. Lines 331-332: in the sentence "...practices rooted in spiritual practice, such as mindfulness, meditation, and yoga, are recommended to prevent burnout", additional references would be required to justify this assertion. Alternatively, the authors may wish to amend the text to read "are sometimes recommended to prevent burnout".
--	--

VERSION 1 – AUTHOR RESPONSE

Reviewer: 1

Dr. LUIS ÁNGEL PÉRULA DE TORRES, Instituto Maimonides de Investigacion Biomedica de Cordoba, Teaching Unit of Family and Community Medicine. Health District of Cordoba and Guadalquivir.

Comments to the Author:

This systematic review asks whether there is a measured association between burnout and spiritual health amongst doctors. This is a qualitative systematic review, since there was no possibility of performing a meta-analysis According to the authors, mindfulness would also be included in spiritual health definitions, and both are promoted as approaches to reduce burnout in the workplace. There

are numerous studies that evaluate the effect of mindfulness on doctor burnout, and these should have been taken into account in the systematic review.

Thank you very much for taking the time to review our article. Mindfulness is not always included in spiritual health definitions. As a practice, mindfulness has its roots with Buddhist theory, but is sometimes used now in mental health provision, and/or as a non-spiritual practice. In our search strategy, we included all major religious practices, on the advice of an information scientist, to provide a comprehensive search that would catch all studies with a quantification of spiritual health. Mindfulness was included on the same basis as other religious practice terms. 'Mindfulness' was not quantified in any of the studies found that also measured burnout. Mindfulness **was** included in the search strategy to pick up any and all studies which had a measure of spiritual health, as it was hypothesised that where mindfulness interventions or similar were being assessed or evaluated, this may have included a spiritual health, wellbeing or distress measure (which would fit the eligibility criteria). A different systematic review could look at the effect of mindfulness on doctor burnout, such a review would be useful in looking at the impact of mindfulness practice or interventions on burnout rates, as reviewed here <https://onlinelibrary.wiley.com/doi/full/10.1111/medu.14020> with no reference to spiritual health-. These studies may measure burnout. However, we can say with reasonable confidence that these studies don't measure spiritual health- if they did, we would have included them, as this systematic review is looking for any study that included medical doctors, and measured burnout **and** spiritual health (or related concepts). Where mindfulness was included, it was as part of spiritual practices, or spiritual routines. "spiritual wellness, spiritual wellbeing" has been added to line 138 to make the focus on spiritual health more clear. We feel confident with our searches that if any studies that measured mindfulness, or participation, and also measured burnout, we would have included those studies. "Search terms were deliberately comprehensive, and related or similar terms to burnout or spiritual health were included in the searches, for example compassion fatigue, moral injury, mindfulness etc. Studies were included if we felt that the study measured any phenomenon very close to occupational burnout, **and** any measure of any aspect of wellbeing close to spiritual health." has been added to the methods. Page 6.

Although this systematic review asks whether there is a measured association between burnout and spiritual health amongst doctors (for which they look for studies with a cross-sectional design), that is, they do not try to prove the existence of a causal relationship, it would have been much more interesting to study this question and thus be able to provide evidence on whether spiritual health is a protective factor against burnout, ideally through experimental studies.

In our eligibility criteria, we did not specify cross-sectional studies only, but these were the only studies found. An experimental study would give higher quality evidence, and could provide evidence of a causal relationship. However, given the topic, it is anticipated there would be difficulties randomising participants to one spiritual path or religion, or none, over another. An experimental study would be an interesting follow on from our review, if it were possible. We have added " , of any design," to line 111.

One study with a very small sample size (n=7) was included in the review, probably providing hardly any valid information.

We agree, 7 is a very small sample size, especially as it only included 4 doctors. This study, Das et al (2016), was a pilot for the Baruah et al (2019) study. Our protocol didn't specify a minimum sample size, and therefore it was felt best to present the study, bearing in mind its small sample, and limited conclusions that could be drawn. We agree it probably adds little, however, given there are few systematic reviews of this topic, we opted not to exclude based on sample size.

The conclusions that can be reached after this systematic review are few and of low added value for clinical practice, given the study question posed (perhaps not very ambitious) and the heterogeneous and inconsistent results observed by the authors, although, given the scarcity of existing literature on

this topic provides a view of the state of the issue that may be useful to implement analytical or experimental studies that allow testing hypotheses.

We agree that few conclusions can be drawn on the topic from this review, other than the scarcity of existing literature, but also that the relationship between spiritual health and burnout is a common hypothesis, with an unclear answer. We agree that further research will be beneficial to test these hypotheses, which we suggest in lines 336-342.

In the discussion it is stated that "one high quality study on the topic found that spirituality routines lower the risk of burnout" and that conclusion of causality cannot be reached with a cross-sectional design such as the one mentioned.

In the article written by Roslan et al (2021), they do state: "We found that regular spirituality routines lowered the risk of developing personal- and work-related burnout among interns." However, we agree that that as cross-sectional study, this claim of causality isn't proven by the study. We have amended the wording in our paper to say: "The one high quality study on the topic stated that spirituality routines lower the risk of burnout, 56 as they found that regular spiritual routines were associated with lower burnout risk, however, we cannot infer causation from the study design." Lines 314-316

Reviewer: 2

Dr. Francesco Chirico, Università Cattolica del Sacro Cuore Facoltà di Medicina e Chirurgia

Comments to the Author:

The PICO strategy and the methods are not sufficiently described to publish this paper as systematic review

We have expanded the methods to provide the level of detail necessary.

Reviewer: 3

Dr. Stephanie Harris, AdventHealth

Comments to the Author:

This paper is well done, fills a gap in the literature, and is methodologically sound.

We thank the reviewer for this assessment of the paper.

Please address in the narrative whether the retrieval was limited to cross-sectional studies or whether that was the only study design retrieved, i.e. what was the rationale for including only this study design. I see that all study designs were included in the search strategy, but this should be stated in the text as well.

Thank you. We have added " Any study design was eligible, but commentaries and editorials were excluded." To line 131.

Additionally, briefly touch on your population of interest and who your strategy included, i.e. did you include resident physicians and fellows? Physicians in all settings, specialties etc. Again, this is in the search strategy, but would be helpful in the narrative.

All medical doctors were included, whatever that meant in the country of the study- i.e. any post-graduate from medical school working in a clinical setting as a doctor. Throughout the manuscript, the term 'doctors' has been changed to 'medical doctors' to make the population of interest clear. The inclusion criteria (lines 128-135) now reads:

“Eligible studies were those reporting quantified evidence about both burnout (or similar concepts including occupational stress, moral injury from work, occupational compassion fatigue etc) and spiritual health (or concepts that included this aspect of health). These data must be reported in a study population that included medical doctors from any speciality or in post-graduate training. Any study design was eligible, but commentaries and editorials were excluded. The WHO definition for burnout¹ and the Whitehead et al² definitions were referred to and considered by two reviewers at screening stage. There were no language restrictions to the searches.”

The background/introduction should be revised to include the following additional points of discussion: the distinction/overlap between religion and spirituality,

We realise that this is subject to some debate, and some cultural differences. As this review considers literature worldwide, we include spiritual health as encompassing all faiths and none. We have added: “Some authors argue that religiosity and spirituality are one and the same⁵, whereas English GPs defined spiritual health as including, but not exclusive to, religion,² and there is a growth in the population that consider themselves ‘spiritual but not religious.’⁶ Therefore, we consider spiritual health to include religious practice and belief, but also to include non-religious spirituality.” To the introduction. (lines 102-106)

2) etiology and ramifications of burnout in physicians (what contributes to burnout and why is burnout in physicians important?),

This is a huge topic that is, in our opinion, under-researched. The aetiology of burnout is likely multifactorial, and it has massive bio-psycho-social-spiritual effects on doctors’ lives. We have been researching the aetiology and effects of burnout on doctors, and have some depth interview data to be published shortly. We have added: “Burnout leads to retirement and resignation⁷ which adds to the workforce crisis, as well as patient safety concerns.⁸ The Covid-19 pandemic has presented huge challenges to the health care workforce.^{9 10} Burnout appears to have a malign effect on all aspects of health,^{2 11} as well as being impacted itself by all aspects of health.¹² The aetiology of burnout is still not fully understood, is likely multi-factorial, and merits further research.¹³⁻¹⁵” to the text. (lines 85-89)

and 3) what are some of the related concepts to burnout (i.e. compassion fatigue) that were searched and what, generally, is their relationship to burnout.

As a systematic review, we wished to gather all evidence from studies that considered a quantification of burnout, and a quantification of spiritual health. We therefore deliberately searched widely, including terms such as compassion fatigue and moral injury, which are considered by some to be similar or related phenomena to occupational burnout. Our goal in this systematic review is not to consider whether burnout, compassion fatigue and moral injury describe the same, similar, or related phenomena. We have added: “Search terms were deliberately comprehensive, and related or similar terms to burnout or spiritual health were included in the searches, for example compassion fatigue, moral injury, mindfulness etc. Studies were included if we felt that the study measured any phenomenon very close to occupational burnout, **and** any measure of any aspect of wellbeing close to spiritual health.” to the methods section of the text. (lines 144-147)

One issue with this paper is the addition of moral injury into the introduction and the search strategy. Though somewhat related to burnout, it is a distinct construct with different contributing factors, and outcomes and it should not be conflated with burnout. The inclusion of this concept is somewhat problematic. As such, the relationship between moral injury, burnout, and spirituality should be explored in the introduction and in the discussion. This can also be addressed in the limitations.

Moral injury is another important concept to consider in the discourse around burnout. This concept refers to the harm caused when someone is required to act contrary to their internal ethical code. Some equate moral injury and burnout, whilst others treat the two as separate concepts. Even without this consensus, it is reasonable to assume that moral injury is implicated in burnout, and for the purpose of this work, we consider the two concepts related, but not necessarily equivalent. When GPs (general practitioners) in the UK were asked to define ‘spiritual health’, being true to an ethical code

was a common thread. Moral injury is mentioned as a thread through which burnout and spiritual health may be related. We have moved and amended the sentence regarding moral injury “Burnout has been associated with moral injury,¹⁶ referring to the harm caused when someone is required to act contrary to their internal ethical code.¹⁷” (line 100-101) to the section discussing the relationship with spiritual health and being true to an ethical code, to try and make this nod to the term clear, without getting in to the contentious debate regarding the overlap between moral injury and burnout.

In addition, the authors should address the issue that religion/spirituality and spiritual well-being/health are not synonymous. Religion/spirituality may be used as a negative coping strategy. It would be helpful to distinguish those papers that distinctly address spiritual well-being versus spiritual/religious observance.

Similarly to the distinction between burnout, compassion fatigue, and moral injury, we have tried to ensure our searches for any spiritual health related phenomena were comprehensive. We don't make any claims that these terms are synonymous, but that we wished to include any kind of measure (including dichotomous 'yes/no' answers) of spiritual health. We recognise that there can be negative spiritual coping, and this may be the reason why two studies showed higher distress in those with higher religiosity. We have added “A number of studies found no association, with two describing greater distress with higher levels of religiosity, perhaps due to use of religious coping as a negative strategy.” to the discussion, line 310.

We are aware of debate around the topics of both burnout, and spiritual health, regarding definitions, and which related phenomena each term encompasses. We take an epistemologically pragmatic approach, and thus have tried to avoid too much contentious debate about the exact boundaries of burnout and/ or spiritual health. We have added a text box to top of the document to try and clarify our position on definitions of the terms burnout and spiritual health.

Lines 105-106 unclear. Does this mean that the papers retrieved directly from authors were not peer-reviewed and thus considered grey literature? Please clarify this statement and the statement about publication bias. Additionally, was any other grey literature searched?

These were studies linked to abstracts found via databases, but the full study details were accessed from the authors, we considered these grey literature because they were not published in peer-reviewed journals, but were still subject to a study appraisal as part of the review methods. No other grey literature was searched. Contact with authors in this area has shown some publication bias for the topic, in that they had received multiple rejections, and also that where a negative or no association was found, the full data were not given, leading to availability bias. We have added reference to availability bias throughout the paper. We have clarified these lines to: “Emails to authors suggested anecdotally that there could be availability and publication bias, and therefore unpublished studies, found via conference abstracts, were included and authors were contacted for more information where limited data were presented. No other grey literature was included.” (lines 152-155)

Overall, the authors conducted a thorough systematic review and this is a valuable addition to the literature.

We are grateful for this assessment of our paper

Reviewer: 4
Dr. Edward Spilg, University of Ottawa

Comments to the Author:
Overall, this is a well written manuscript which addresses an important gap in the literature.

We are grateful to the reviewer for these helpful comments.

Abstract: This is clear and appropriately informative.
The authors should specify in lines 29, 32, 34 (and throughout the text) which type of "doctors" ie medical doctors or consider using the term physicians to avoid confusion with other types of "doctors"

given the term "doctors" is more widely applied to healthcare professionals other than physicians in North America than it is in the UK.

Thank you for highlighting this. - As physicians is a specific term used in the UK and elsewhere for adult internal medicine doctors, we have changed all uses of the term 'doctors' to 'medical doctors'.

Introduction: This clearly sets out the rationale and justification for the systematic review. This appears well written.

Line 89: suggest change term doctors to physicians.

As above, we have changed the term to 'medical doctors' to clarify for an international audience.

Methods: As a non-expert in systematic review methodology, this section appears appropriate but I would recommend it is checked by an expert in systematic review methodology.

Line 96: suggest change term doctors to physicians.

As above, we've changed the term to 'medical doctors'.

Results: The results are clear and stated in a systematic fashion and focus on the study objectives.

As a non-expert in systematic review methodology, I would recommend this section is checked by an expert in systematic review methodology.

Lines 133, 134, 139, 182: suggest change term doctors to physicians.

Table 2: suggest change term doctors to physicians throughout for consistency.

In Table 2, we've reflected the terms used within the studies- i.e. if the studies referred to doctors, nurses, etc, or physicians, or more specific specialities, we have used those terms.

Discussion: This section, I believe, is well written overall.

Thank you

Lines 252, 260, 261, 264, 274, 326: suggest change term doctors to physicians.

As above, changed the term to 'medical doctors'.

Line 256-257: The statement "with a few studies from more secular areas" needs clarification, especially why the USA is not considered a "secular area" for this purposes of this study.

The USA contrasts markedly with Northern Europe in terms of religiosity.

<https://www.pewresearch.org/fact-tank/2019/05/01/with-high-levels-of-prayer-u-s-is-an-outlier-among-wealthy-nations/>

We have added: "(The USA has high levels of religiosity^{18 19} than, for example, many northern European countries.²⁰)" to the text to explain why we consider other areas more secular than the USA.

Line 269 and 289: should replace "residents" with "resident physicians"

While we understand the need for clarity, in discussion of Chow et al, we have again reflected the language used by the authors- they do not clarify that all those considered in their review are resident physicians, simply that they are in residency training. Similarly, Purvis et al refer to 'residents', and as the study setting is a critical care unit, these could be resident critical care doctors, rather than physicians? We have left this as residents, because this reflects the source data.

Conclusion: The conclusions are clear and relate to the findings in the text.

Thank you

Lines 329, 336: suggest change term doctors to physicians.

As above, used the term 'medical doctors'.

Lines 331-332: in the sentence "...practices rooted in spiritual practice, such as mindfulness, meditation, and yoga, are recommended to prevent burnout", additional references would be required to justify this assertion. Alternatively, the authors may wish to amend the text to read "are sometimes recommended to prevent burnout".

Mindfulness, meditation and yoga are often recommended for those concerned about burnout. "Often" has been added to the sentence, and two more references have been added where these things are recommended for burnout.

VERSION 2 – REVIEW

REVIEWER	LUIS ÁNGEL PÉRULA DE TORRES Instituto Maimonides de Investigacion Biomedica de Cordoba
REVIEW RETURNED	22-Mar-2023

GENERAL COMMENTS	Systematic review that addresses a topic of interest and relevance, the study question being pertinent. One of the limitations of this narrative systematic review is that it focuses on showing the evidence of the relationship or association between burnout and spiritual health, or vice versa, selecting observational, cross-sectional studies, where the causal relationship is not possible to establish. The ideal would have been to search for, collect and analyze analytical or experimental observational studies. This limitation should be considered in the article. The bibliographical references used in the introduction on the effectiveness of mindfulness in reducing burnout (20 and 21) are not very current (2012 and 2016) or the main ones. There are other published studies (where self-pity is also studied as a protective factor) that are newer, more representative or more recent, such as these:  -Kabat-Zinn J, Torrijos F, Skillings AH, Blacker M, Mumford GT, Levi-Alvares D, Santorelli S, Rosal MC. Delivery and effectiveness of a dual language (English/Spanish). Mindfulness-based stress reduction (MBI). Program in the Inner City -a seven-year experience: 1992-1999. Mindfulness Compassion.2016;1:2–13. https://doi.org/10.1016/j.mincom.2016.09.007. -Rosa Magallón-Botaya, Luis Angel Pérula-de Torres, Juan Carlos Verdes-Montenegro Atalaya, Celia Pérula-Jiménez, Norberto Lietor Villajos, Cruz Bartolomé-Moreno, Javier Garcia-Campayo, Herminia Moreno-Martos & the Minduudd Collaborative Study Group. Mindfulness in primary care healthcare and teaching professionals and its relationship with stress at work: a multicentric cross-sectional study. BMC Family Practice 2021; 20 (1): 24. doi: 10.1186/s12875-019-0913-z. In the methodology, emotional health or similar terms such as exposure and burnout or other similar concepts such as outcome have been considered, but since the study designs were cross-sectional, the possible cause-effect relationship could be reversed. that is to say, that it was a matter of answering the inverse question: subjects with or without burnout (exposure), present greater or lesser spiritual health (outcome)?.  -The referenced citation numbers must all be in superscript format and without parentheses. -Table 1: some results are not clear. For example, in the study by Clark (citation 8, 2007), it is concluded that higher job satisfaction was associated with higher spirituality, but the p value does not seem to be statistically significant (p 0.433). In the study by Frank (citation 14, 1999), it is stated that there is no association, when the value of the Odds ratio was 1.3 (1.1-1.6), and since the confidence interval does not include the value 1, it should be interpreted as that if it would have a statistical relationship, even though the magnitude of the effect is small. In the study by Glasberg et al (citation 15, 2007), the estimator or statistical test used is not provided, only the p value. The same is true of the Purviss study (citation 31, 2019). -Figures 2 and 3 may be dispensable, since they provide little information, summarizing the results in the text.
--

REVIEWER	Stephanie Harris AdventHealth, Center for Nursing, Whole-Person, and Academic Research
REVIEW RETURNED	28-Mar-2023

GENERAL COMMENTS	Overall, this is a well-executed and well-written review. I appreciate the authors addressing initial concerns. The title of the paper should reflect that related concepts are included in this systematic review. As stated in the initial review of the paper, although searched concepts are related to burnout, they are also distinct and change the scope of this outcome. For example, moral injury is a psycho-spiritual injury at its core; thus, spiritual health is patently associated, whereas burnout is more of a chronic, cumulative occupational stress injury. In addition, it is suggested that the authors address how and why they selected concepts and terms related to burnout. For example, why the inclusion of moral injury but not moral distress or secondary/vicarious trauma? And although the authors state there is no consensus regarding etiology of burnout, related concepts like work-schedule tolerance are relevant. Framing the strategy for selecting related concepts would help the reader to better understand the methodology and the scope of the paper. Lastly - small edit - it looks like there is a word missing in line 192 - a descriptor before the last instance of the word bias.
---

VERSION 2 – AUTHOR RESPONSE

Reviewer: 1

Dr. LUIS ÁNGEL PÉRULA DE TORRES, Instituto Maimonides de Investigacion Biomedica de Cordoba, Teaching Unit of Family and Community Medicine. Health District of Cordoba and Guadalquivir.

Comments to the Author:

Systematic review that addresses a topic of interest and relevance, the study question being pertinent. One of the limitations of this narrative systematic review is that it focuses on showing the evidence of the relationship or association between burnout and spiritual health, or vice versa, selecting observational, cross-sectional studies, where the causal relationship is not possible to establish. The ideal would have been to search for, collect and analyze analytical or experimental observational studies. This limitation should be considered in the article.

We have added: “Only observational, cross-sectional studies, were found, and therefore we were unable to analyse whether there is any causal relationship between spiritual health and burnout” to the text, as there were no experimental studies.

The bibliographical references used in the introduction on the effectiveness of mindfulness in reducing burnout (20 and 21) are not very current (2012 and 2016) or the main ones. There are other published studies (where self-pity is also studied as a protective factor) that are newer, more representative or more recent, such as these:

-Kabat-Zinn J, Torrijos F, Skillings AH, Blacker M, Mumford GT, Levi-Alvares D, Santorelli S, Rosal MC. Delivery and effectiveness of a dual language (English/Spanish). Mindfulness-based stress reduction (MBI). Program in the Inner City -a seven-year experience: 1992-1999. Mindfulness Compassion.2016;1:2–13. <https://doi.org/10.1016/j.mincom.2016.09.007>.

Thank you for drawing our attention to this interesting study. However, it does not appear to involve medical doctors, or their burnout.

-Rosa Magallón-Botaya, Luis Angel Pérula-de Torres, Juan Carlos Verdes-Montenegro Atalaya, Celia Pérula-Jiménez, Norberto Lietor Villajos, Cruz Bartolomé-Moreno, Javier Garcia-Campayo, Herminia Moreno-Martos & the Minduudd Collaborative Study Group. Mindfulness in primary care healthcare and teaching professionals and its relationship with stress at work: a multicentric cross-sectional study. BMC Family Practice 2021: 20 (1): 24. doi: 10.1186/s12875-019-0913-z.

Thank you for drawing our attention to this interesting planned study. I'll be really interested to read the findings.

In the methodology, emotional health or similar terms such as exposure and burnout or other similar concepts such as outcome have been considered, but since the study designs were cross-sectional, the possible cause-effect relationship could be reversed. that is to say, that it was a matter of answering the inverse question: subjects with or without burnout (exposure), present greater or lesser spiritual health (outcome)?

We agree. As above, we've added "Only observational, cross-sectional studies, were found, and therefore we were unable to analyse whether there is any causal relationship between spiritual health and burnout" to the strengths and weaknesses section of the discussion.

-The referenced citation numbers must all be in superscript format and without parentheses.

I will ask for this to be clarified, as the opposite request has been made in previous reviews.

-Table 1: some results are not clear. For example, in the study by Clark (citation 8, 2007), it is concluded that higher job satisfaction was associated with higher spirituality, but the p value does not seem to be statistically significant (p 0.433).

Presenting the raw statistical data in a conventional way that conveys meaning with the heterogeneity of methods has been a challenge in this review. This paper compares structural path models to explain their finding that spirituality, integration and self-actualization explained 22% of the variation in job satisfaction ($R = 0.48$; adjusted $R^2 = 0.218$; $df = 3,175$; $F = 17.2$; $p = 0.001$)- We have added this data to table 1 to try and make the summarised authors' conclusions more congruent and understandable.

In the study by Frank (citation 14, 1999), it is stated that there is no association, when the value of the Odds ratio was 1.3 (1.1-1.6), and since the confidence interval does not include the value 1, it should be interpreted as that if it would have a statistical relationship, even though the magnitude of the effect is small.

This paper found a small association between religious fervour, and desire to become a physician again- in analysis, we didn't feel this added much data either way as to whether burnout and spiritual health were associated. The text has been changed to: "No direct association found between spiritual health and burnout."

In the study by Glasberg et al (citation 15, 2007), the estimator or statistical test used is not provided, only the p value. The same is true of the Purviss study (citation 31, 2019).

Unfortunately, data can only be provided where the authors have given it.

-Figures 2 and 3 may be dispensable, since they provide little information, summarizing the results in the text.

Figures 2 and 3 may be dispensable, but they do give a visual representation of the data in more detail than given in the text.

Reviewer: 3

Dr. Stephanie Harris, AdventHealth

Comments to the Author:

Overall, this is a well-executed and well-written review. I appreciate the authors addressing initial concerns.

The title of the paper should reflect that related concepts are included in this systematic review. As stated in the initial review of the paper, although searched concepts are related to burnout, they are also distinct and change the scope of this outcome. For example, moral injury is a psycho-spiritual injury at its core; thus, spiritual health is patently associated, whereas burnout is more of a chronic, cumulative occupational stress injury. In addition, it is suggested that the authors address how and why they selected concepts and terms related to burnout. For example, why the inclusion of moral injury but not moral distress or secondary/vicarious trauma? And although the authors state there is no consensus regarding etiology of burnout, related concepts like work-schedule tolerance are relevant. Framing the strategy for selecting related concepts would help the reader to better understand the methodology and the scope of the paper.

The overlap between burn out, moral injury, secondary trauma (also know as compassion fatigue, which we included) etc are worthy of much discussion! We have sought to balance the need for clarity of the phenomena under investigation, with the need for a broad and thorough interrogation of the evidence. The search strategy was deliberately broad, however our analysis and focus for the review has been any association between the concepts of burnout and spiritual health in medical doctors. The definitions used for burnout and spiritual health in the analysis have been added in a text box at the previous review stage, to provide clarity as to the scope of the paper..

Lastly - small edit - it looks like there is a word missing in line 192 - a descriptor before the last instance of the word bias.

This has been clarified to be information bias.